# Extracellular Vesicles Enhance Multiple Myeloma Metastatic Dissemination

**DOI:** 10.3390/ijms20133236

**Published:** 2019-07-01

**Authors:** Michela Colombo, Domenica Giannandrea, Elena Lesma, Andrea Basile, Raffaella Chiaramonte

**Affiliations:** 1Department of Health Sciences, Università degli Studi di Milano, I-20142 Milano, Italy; 2Department of Oncology and Hemato-Oncology, Università degli Studi di Milano, I-20122 Milano, Italy

**Keywords:** extracellular vesicle, exosome, microvesicle, multiple myeloma, metastatic niche, immune response, mesenchymal cell, osteoclast, osteoblast, angiogenesis

## Abstract

Extracellular vesicles (EVs) represent a heterogeneous group of membranous structures shed by all kinds of cell types, which are released into the surrounding microenvironment or spread to distant sites through the circulation. Therefore, EVs are key mediators of the communication between tumor cells and the surrounding microenvironment or the distant premetastatic niche due to their ability to transport lipids, transcription factors, mRNAs, non-coding regulatory RNAs, and proteins. Multiple myeloma (MM) is a hematological neoplasm that mostly relies on the bone marrow (BM). The BM represents a highly supportive niche for myeloma establishment and diffusion during the formation of distant bone lesions typical of this disease. This review represents a survey of the most recent evidence published on the role played by EVs in supporting MM cells during the multiple steps of metastasis, including travel and uptake at distant premetastatic niches, MM cell engraftment as micrometastasis, and expansion to macrometastasis thanks to EV-induced angiogenesis, release of angiocrine factors, activation of osteolytic activity, and mesenchymal cell support. Finally, we illustrate the first evidence concerning the dual effect of MM-EVs in promoting both anti-tumor immunity and MM immune escape, and the possible modulation operated by pharmacological treatments.

## 1. Biogenesis and Characteristics of Extracellular Vesicles

Extracellular vesicles (EVs) can be released by all kinds of cell types and are found in most biological fluids. They are mainly classified according to different features: biogenesis, size, density, and cargo, which can change depending on EV origin, the overall status of the producing cells, and the surrounding microenvironment.

In the last years, EVs have emerged as key mediators of the pathological interplay between cancer cells and the healthy surrounding cells due to their cargo of lipids, transcription factors, mRNAs, non-coding regulatory RNAs, and proteins [1,2,3].

EV classification is based on their origin and cargo, and allows the identification of three main subgroups: (i) exosomes, vesicles with a diameter below 100–150 nm, deriving from the endocytic compartment; (ii) microvesicles, generated directly by plasma membrane budding and characterized by a wider size range (100–1000 nm); and (iii) apoptotic bodies, big membranous structures (diameter > 2000 nm) generated directly from the cytoplasmic membrane upon activation of the apoptotic cascade [1].

Exosomes arise from intraluminal vesicles (ILVs) contained in late endosomes or multivesicular bodies (MVBs). MVBs containing ILVs may then fuse with lysosomes, forming mature lysosomes, or with the plasma membrane, releasing exosomes [4]. Exosomal cargo is represented by molecules actively and specifically selected by the endosomal sorting complexes required for transport (ESCRT) and loaded into the ILVs for subsequent degradation or recycling. Although exosomal content partially reflects the composition of the producing cells, it is not identical, since it results from the selection of specific molecules [4]. The fusion of MVB with the cytoplasmic membrane and the consequent exosome release are characterized by the activation of proteins involved in MVBs’ docking, such as the actin regulator cortacin, Rab family of GTPases, SNAP receptor (SNARE) proteins, and the fusion regulator synaptotagmin-7. The biogenesis and release of microvesicles is less characterized, but clearly involves different components of the same complexes involved in ILV generation. Variation in content and distribution of lipids that form the plasma membrane may affect the release of microvesicles [5]. Of note, since the current methodologies do not distinguish between exosomes, microvesicles, and apoptotic bodies, in this review we will use the generic term EVs, which includes all the different vesicle subtypes.

EVs can affect the features and functions of receiving cells by delivering many different classes of molecules, such as transcription factors, mRNAs, non-coding regulatory RNAs, and infectious particles. The content of EV partially reflects the cellular origin. Tumor-derived EVs share with EVs of different origins a great number of proteins including adhesion molecules such as tetraspanins and integrins, antigen presenting molecules (MHC class I and II), membrane transport and fusion molecules (annexins, flotillin, and Rab proteins), cytoskeletal proteins (actin, tubulin, and moesin), and many others such as heat shock protein 70 (HSP70) [6]. In addition, they express cell-specific molecules that can often be considered as immunophenotypical markers such as syndecan-1/CD138, a plasma cell marker characteristic of multiple myeloma cells [7].

## 2. Multiple Myeloma Cell Dissemination

Multiple myeloma (MM) is a hematological neoplasm deriving from the clonal proliferation of malignant plasma cells (PCs) [8,9]. MM mostly relies on the tumor microenvironment for its progression. The bone marrow (BM) represents a highly specialized and supportive myeloma niche. Within the BM, PCs take advantage of the local healthy cell populations including mesenchymal stromal cells (MSCs), osteoblasts (OBs), osteoclasts (OCs), endothelial cells, and cells of the immune system, and are sustained by a very supportive milieu rich in cytokines and growth factors [8,9].

Tumor metastasis is the major cause of death in cancer patients. Furthermore, the spread of distant bone lesions is a key event in MM progression. Through a process similar to bone metastases diffusion from primary carcinoma, malignant PCs can recirculate within the blood and finally settle at different sites where they can create new metastatic lesions. The metastatic process is characterized by consecutive steps that include colonization and survival of micrometastasis, dormancy, and finally reactivation and formation of macrometastasis, thereby interfering with physiological bone homeostasis [10].

BM is the most suitable microenvironment for myeloma cell needs. Therefore, it is not surprising that malignant PCs mostly reproduce secondary tumor lesions at distant BM skeletal districts [11]. Nonetheless, the BM of MM patients differs in its cellular and non-cellular composition from that of healthy individuals [12], suggesting that the reprogramming of the distant niche is necessary to favor MM cell homing. It is well known that the choice of metastatization site is determined by the primary tumor ability to precondition distant body districts to allow and support metastatic tumor cell settlement and formation of new tumor lesions. In the following sections we will analyze how EVs released by MM cells (MM-EVs) contribute to this process to favor MM cell dissemination and the formation of new bone lesions.

## 3. EVs Support Myeloma Cell Journey to the Metastatic Site

MM-EVs have been found both in the MM patients’ peripheral blood and BM, and their levels in the blood circulation are positively correlated with the number of bone lesions [13]. Although this is not direct evidence that MM-EVs play a role in tumor dissemination, the analysis of EV content further substantiates this hypothesis. Indeed, EV cargo is enriched with microRNAs (miRNAs) and long non-coding RNAs (lncRNAs), which promote proliferation and osteolysis—two key events that occur during premetastatic niche education and metastasis formation [13,14].

Roccaro et al. showed that exosomes produced by BM-MSCs conditioned by MM disease promote MM cell growth, and are also involved in MM cell dissemination in vivo, a distinct process from normal BM-MSC-derived exosomes [15].

Recently, several groups analyzed the molecular basis of EV pro-metastatic potential, providing a better understanding of their contribution to the different steps of MM cell dissemination and to the development of new lesions. Malignant PCs are initially released from the primary location, protected by an immunosuppressive environment. After leaving the BM, during their journey in the blood stream, myeloma cells are exposed to attacks from the immune system and are subjected to shear stress due to circulation flux. The prevention of tumor metastasis has its major actors in cytotoxic CD8^+^ T cells (CTL), natural killer (NK) cells, and non-classical ‘‘patrolling’’ monocytes [16,17,18]. Recent findings from Nielsen and colleagues [19] showed that MM-EVs favor MM cell diffusion to distant skeletal sites thanks to their procoagulant activity (Figure 1).

Indeed, microvesicles from MM patients exert a procoagulant activity due to both tissue factor (TF) and procoagulant phospholipids (PPLs), resulting in thrombin generation [19]. Thrombin triggers platelet activation, polymerizes fibrinogen to fibrin and triggers its crosslinking [20]. The interaction between platelet/fibrin and tumor cells promotes metastasis formation in different ways: (i) by providing a cover that protects the neoplastic embolus and possibly newly settled micrometastases from NK cell-mediated elimination [20]; (ii) by stimulating the extravasation through the upregulation of adhesion molecules and the increase in endothelial permeability [21,22,23]; and (iii) by preconditioning the “metastatic niche” with platelet-derived cytokines such as TGF-β [21]. In MM patients, platelet activation correlates with disease progression. The crucial role of platelets in MM cell dissemination is further supported by evidence that MM cells preconditioned with platelets showed significantly increased tumor cell engraftment ability in vivo, with a mechanisms dependent on IL-1β release [24]. This cytokine has a pleiotropic effect on MM cell proliferation, in addition to the potential to activate OCs and the expression of adhesion molecules for extravasation such as integrin α4β1 and CD44 [25,26].

## 4. EV Uptake and Cargo Delivery to the Premetastatic Niche

A second important step in promoting metastasis formation is EV ability to reach the premetastatic niche and deliver pro-tumor signals to the local microenvironment. EVs are highly enriched in cholesterol, annexin, sphingomyelin, glycosphingolipids, and phosphatidylserine, which make them stable structures released in the interstitial space or into the biological fluids such as blood, thereby acting on short or long distances and promoting autocrine and paracrine signaling [27].

EV tropism is a property essential to transfer EV cargo to selected recipient cells at distant BM sites. Increasing lines of evidence suggest that tumor-derived EVs might be uptaken by organ-specific cells where they can participate in preconditioning the premetastatic niche [28,29].

EV uptake by recipient cells is energy dependent [30]. Several lines of evidence support the hypothesis that EVs are mostly internalized through clathrin-mediated and caveolin-dependent endocytosis, at least in part, promoted by ligand/receptor interactions. This interplay seems to be a cellular-specific mechanism and may lead to the activation of different downstream biological processes such as signaling transduction, endocytosis, and membrane fusion. Macropinocytosis and phagocytosis appear to be less frequent [31]. EV uptake may depend on the lipid composition (in particular, changes in the organization of lipid rafts can modify EV propensity to fuse), and an acidic microenvironment seems to promote membrane fusion events [27].

Several potential receptors favoring exosome uptake have been identified. These include Tim1/4 for B cells [32] and ICAM-1 for antigen presenting cells (APCs) [33], while heparan sulfate proteoglycans have been reported as receptors for cancer cell-derived exosomes [34]. Syndecan-1 (CD138) is a cell surface heparan sulfate-bearing proteoglycan expressed at high levels on MM cells for which it also represents a selective marker. Syndecan-1 activated by high levels of heparanase in MM cells promotes the spontaneous metastasis of myeloma cells to the bone [35]. A possible explanation is supported by syndecan-1 expressed on exosomes, which promotes their uptake by BM-MSCs with a mechanism based on the ability of fibronectin to bind to heparin sulfate molecules on the membrane surfaces of both MM-EVs and BM-MSCs [36] (Figure 1).

## 5. MM-EVs Promote Angiogenesis and Production of Angiocrine Factors

Angiogenesis is a key process in metastasis formation. Besides promoting tumor cell growth at the primary site and favoring the dissemination of neoplastic cells in the blood circulation [37], angiogenesis plays a crucial role also in the preparation of the premetastatic niche. Indeed, the high permeability and leakiness of the newly formed vessels at the metastatic niche favor cancer cell extravasation from the circulation. Moreover, endothelial cells of new vessels may release angiocrine factors that nurture tumor cells when reaching the premetastatic niche, thereby supporting the survival and growth of disseminated tumor cells [38,39]. Nonetheless, a full angiogenic switch is necessary for the progression of dormant or micrometastatic tumors to macrometastases [40].

Tumor-derived EVs are known to participate in neoangiogenesis and promote endothelial branching both locally and at distant sites. EVs endowed by angiogenic properties are shed by cells from different types of tumor, including breast cancer, pancreatic carcinoma, glioblastoma, bladder cancer, chronic myelogenous leukemia, etc. [41].

Although a complete characterization of MM-EV content remains unavailable, the presence of several angiogenic factors has been reported, including VEGF, angiogenin, bFGF, Serpin E1, TIMP-1, and PDGF [42,43]. Accordingly, as shown in Figure 2, MM-EVs can induce endothelial cell proliferation, tube formation, and new vessel development through the modulation of different pathways such as c-Jun N-terminal kinase (JNK), signal transducer and activator of transcription 3 (STAT3), and p53 [42,43], in addition to promoting the secretion of angiocrine molecules such as VEGF, ICAM-1, and IL-6 by endothelial cells [44,45].

Further studies indicate that components of MM-EV cargo such as CD147 [46], piRNA-823 [45], and miR-135b [47,48] may also contribute to the angiogenic effects of MM-EVs. In particular, Umezu et al. demonstrated that exosomal miR-135b directly promotes endothelial tube formation by blocking the expression of factor-inhibiting hypoxia-inducible factor 1 (FIH-1) in endothelial cells [47,48]. Finally, besides its role in exosome uptake, MM cell marker syndecan-1/CD138, carried by MM-EVs, plays a key role in promoting endothelial cell invasion and angiogenesis [49]. This function is consistent with the correlation of syndecan-1 gene expression and BM microvessel density in patients with monoclonal gammopathy of uncertain significance (MGUS) and MM [50]. It remains to be investigated if these mechanisms influence the formation of new vessels at the premetastatic niche.

Interestingly, MM-EVs treated with bortezomib or lenalidomide exhibit decreased ability to activate NF-κB and to induce VEGF, IL-6, and bFGF expression in endothelial cells, thus showing decreased angiogenic potential [43,51]. This suggests that these two chemotherapeutics may counteract MM progression by affecting EV-mediated communication between MM and endothelial cells.

## 6. MM-EVs Promote the Formation of New Bone Lesions

Bone resorption regulates different aspects of metastasis formation, from the creation of space in the osseous matrix for tumor cell settlement to the release of entrapped pro-tumor factors, including TGF-β and insulin-like growth factor (IGF) [52]. OCs account for 1–4% of the total cells in bone and are responsible for its resorption. In MM bone disease, monocytes [52] and myeloid derived suppressor cells (MDSCs) are OC sources [53]. During MM progression, the functional balance between OCs and OBs (cells of mesenchymal origin which synthesize the bone matrix) is definitively perturbed, with an increase in OC lytic activity [13].

As illustrated in Figure 3, the first indication of a potential role of MM-EVs in promoting the dissemination of myeloma bone disease was reported by Zhang et al. who identified a correlation between peripheral blood CD138^+^ circulating EVs and bone lesions in de novo MM patients [13].

Raimondi et al. also showed that MM-derived exosomes directly support bone osteolysis by favoring the migration and differentiation of OC precursors [54]. The authors performed in vitro studies to demonstrate that MM-derived exosomes contribute to OC migration, by increasing CXC chemokine receptor type 4 (CXCR4) expression, and promote OC viability by enhancing the anti-apoptotic gene expression of Bcl-2, survivin, and AKT phosphorylation [54]. Finally, MM-derived exosomes regulate OC bone resorbing activity by increasing the expression of key osteoclastogenic enzymes, i.e., tartrate-resistant acid phosphatase (TRAP), cathepsin K (CTSK), and matrix metallopeptidase 9 (MMP9) [54].

Interestingly, external stimuli, such as chemotherapeutic treatments or hypoxia, may increase the osteoclastogenic potential of MM-EVs. Indeed, treatments with standard-of-care drugs such as bortezomib, carfilzomib, and melphalan increase the levels of heparanase in MM-derived exosomes [55]. Heparanase activates the ERK pathway resulting in MM cells releasing the osteolytic factors, MMP-9 and receptor activator of nuclear factor-κB ligand (RANKL). Not surprisingly, high levels of heparanase stimulate systemic osteoclastogenesis and osteolysis [56]. These lines of evidence suggest that drugs currently used in MM therapy to inhibit tumor growth and bone disease may contemporaneously ignite a burst of MM-EVs carrying high levels of heparanase, which can eventually contribute to promoting OCs differentiation. OCs in turn support MM cell growth and survival through the secretion of IL-6 and osteopontin, finally favoring patients’ relapse [57].

Hypoxic BM (1–2% O2) conditions cause an increase in IL-32 expression in MM cells, which is associated with poor survival and bone loss. Zahoor and colleagues reported that the inflammatory IL-32 can be delivered through MM-EVs, and plays a key role in promoting OC activity [58]. More recently, Alessandro et al. [59] identified another protein delivered by MM-derived exosomes involved in osteoclastogenesis, epithelial growth factor receptor (EGFR) ligand amphiregulin (AREG). AREG leads to the activation of EGFR in OC precursors, directly inducing their differentiation into mature OCs. Moreover, AREG blocks MSC osteogenic differentiation and induces the release of the osteoclastogenic cytokine IL-8. MM-derived exosomes further induce an imbalance between OCs and OBs by reducing the MSC-mediated release of osteoprotegerin (OPG), and increasing that of RANKL in an AREG-independent manner [59].

The evidence that MM-EVs not only support the formation of bone lesions by enhancing OC activity, but also by inhibiting OB differentiation has been confirmed by several groups. Zhang et al. demonstrated that MM-EVs inhibit osteogenesis of BM-mesenchymal stem cells in vitro and exacerbate myeloma bone diseases in vivo [13]. Faict et al. showed that MM-EVs in the murine model 5TGM1 cause the downregulation of Runt-related transcription factor 2 (RUNX2) and the Wingless-related integration site (WNT) pathway, which positively regulate OB differentiation by transferring inhibitors of the osteogenic pathways such as Dickkopf-1 (DKK-1) [60]. Furthermore, Li and colleagues showed that RUNX2 can also be suppressed by MM-derived exosomes through the transfer of the long non-coding RNA lncRUNX2-AS1 in mesenchymal stem cells. IN this context, lncRUNX2-AS1 interferes with the splicing of RUNX2 repressing osteogenesis [61]. Of note, increasing evidence supports the hypothesis that lncRNAs may represent a significant component of MM-EV cargo, and may have a role in the communication between MM cells and the surrounding niche [14,62].

According to recent studies, miRNAs carried by cancer cell-derived EVs, i.e., miR-192 [63] and miR-940 [64], also contribute to the OC–OB imbalance by promoting the differentiation of one cell population at the expenses of the other [65]. Concerning MM, it has been reported that miR-21 enhances the STAT3-dependent signaling by inhibiting PIAS3, thereby resulting in OC differentiation [66]. The miR-21 passenger strand (miR-21-3p), along with miR-103a-3p and miR-181a-3p involved in OB or OC’ differentiation, have been found in EVs shed by the myeloma RPMI8226 cell line together with protein regulators of bone lesions such as DKK-1, IL-7, and sFRP2 (secreted Frizzled Related Protein 2) [67,68].

## 7. MM-EVs Educate Bone Marrow Mesenchymal Cells

BM-MSCs play a major role in regulating micrometastasis growth, and MM cells have been reported to educate BM-MSCs inducing a pro-tumor phenotype, thus resulting in the release of factors that promote tumor survival, proliferation, migration [69], and osteoclastogenesis [70,71]. As illustrated in Figure 4, EVs shed from MM cells and BM-MSCs are key players regulating micrometastasis growth.

Recent work indicates that normal BM-MSC-derived exosomes (MSC-exosomes) display a tumor suppressive potential due to the presence of tumor suppressive miRNAs, such as miR-15av, largely consistent with their ability to promote tumor cell dormancy upon the initial micrometastasis settlement [15]. A confirmation of the growth-inhibitory behavior of MSC-EVs has been reported in bone metastatic breast cancer that shares the same BM microenvironment as MM and is in contact with the same healthy MSC-EVs. Healthy MSC-EVs induce breast cancer cells to enter the G0/G1 phase by upregulating tumor suppressive miRNAs such as miR-127, miR-197, miR-222, and miR-223 [72]. Similarly, miR-23b carried by EVs from BM mesenchymal stem cells of a human donor can induce tumor cell dormancy in a bone marrow-metastatic human breast cancer cell line by suppressing a gene, myristoylated alanine-rich C-kinase substrate (MARCKS), encoding a protein that promotes cell proliferation and motility [73]. Although this tumor growth inhibitory mechanism activated by MSC-EVs has not been studied in MM progression, reasonably it might also affect MM cell growth. Indeed, miR-23b [74], miR-197 [75], and miR-223 [76], also play a tumor suppressor role in MM by negatively regulating key oncogenes such as Mcl-1 and Notch2 [75,76].

On the other hand, the switch from micrometastasis to macrometastasis may be favored by MSC preconditioning which promotes a pro-tumor activity of MSC-EVs. MM-EVs precondition BM-MSC by promoting their viability via STAT3 and JNK phosphorylation [42].

Moreover, miRNAs carried by EVs derived from the MM cell line OPM-2—miR-146 and miR-21—stimulate cytokine production in mesenchymal stem cells and their transformation into carcinoma-associated fibroblasts (CAFs) [77]. In particular, changes induced by MM-EV-derived miR-146a in BM-MSCs include Notch1 activation and the release of cytokines, in addition to chemokines such as IL-6, IL-8, CXCL1, IP-10, CCL2, and CCL5, which favor MM cell growth, viability, and migration [78,79].

As a result of MM-EV preconditioning, BM-MSCs can produce EVs (MM MSC-EVs). Normal and MM BM-MSC-derived exosomes (MM MSC-exosomes) display different properties. In fact, while the former hamper MM cell growth, MM MSC-exosomes are able to promote MM cell proliferation and dissemination, due to the selective transfer of key cytokines, such as IL-6 and CCL2, or junction plakoglobin and fibronectin that may increase MM cell adhesive ability [15].

## 8. The Dual Effect of MM-EVs in Anti-Tumor Immunity and Immune Escape

The ability of tumor cells to escape the immune system control is a key event in tumor progression. During the metastatic process, tumor cells leave the immunosuppressed primary site to reach the metastatic site. During the metastatic journey and settlement at distant sites, cancer cells are more exposed to the anti-tumor immune system attack [80].

During tumor progression, cancer-derived EVs have a dual role, acting both as immune suppressors or immune promoters. Tumor-derived EVs can trigger the anti-tumor response [81] since they carry tumor antigens that can be taken up by dendritic cells and presented to T cells, thereby triggering their activation [82]. Other immune activating mechanisms rely on EV cytokine content, i.e., IL-15 and IL-18 that promote NK cell cytotoxicity and proliferation [83], or the expression of heat shock protein 70 (Hsp70) able to switch regulatory T cells (T_Reg_) into T helper 17 cells (Th17) and inhibit tumor growth [81].

Several groups have demonstrated the potential role of cancer-derived EVs in suppressing the immune responses against different types of cancers mediated by NK and CD8^+^ T cells [84,85,86]; creating an immunosuppressive pro-tumor environment by expanding the T_Regs_ population [87], stimulating M2 macrophage polarization [88] and dendritic cell tolerance [89,90]; or by suppressing CTL response through PD-L1 delivery [91].

The first observation in multiple myeloma is that MM-EVs are able to carry several pro- and anti-inflammatory cytokine and chemokines, i.e., IL-6, CCL2 [15], IL-15 [83], IL-7 [67], IL-32 [58], IL-10 and IL-16 [92], and VEGF [42]. MM-EVs may also contribute to changing the cytokine milieu, by favoring BM-MSC-mediated release of cytokines and chemokines such as IL-6, CXCL1, IP-10, CCL5 [78], and IL-8 [59]. Although all these cytokines are well known players in inflammatory and immune response, the overall immune modulating effects of MM-EVs in the myeloma setting require clarification.

On the one end, MM-EVs can be potential activators of the anti-tumor immune response since they carry on their surface the same tumor-specific or tumor-associated antigens as the donor MM cells. Indeed, it has been demonstrated that EV-derived tumor antigens can induce the activation of dendritic cells that in turn stimulate the autologous anti-tumor T-cell responses [82]. Leaf and colleagues reported that MM-specific antigens, such as MUC-1 and survivin, can be delivered by MM-EVs to the surrounding APCs, thus inducing their activation. MM-EVs carry the specific immunoglobulin produced by the malignant plasma cells of origin. This represents a unique antigenic determinant (idiotype), which may serve as a tumor-specific antigen [93,94]. Furthermore, Caivano et al. found that EVs from myeloma patients express CD38, a recognized tumor-associated antigen, which is the target of recent immunotherapeutic approaches [95].

Interestingly, the immunogenic activity of MM-EVs (Figure 5A) can be potentiated by inflammatory stimuli or pharmacological treatment.

Indeed, Xie et al. demonstrated that EVs produced by MM cells previously engineered to overexpress TNF-α can enhance the CD8^+^ T lymphocyte response in vivo preventing tumor growth after subcutaneously challenging mice immunized with MM-EVs. These data suggest that inflammatory cytokine stimuli can enhance EV immunogenic activity [96]. Similarly, pharmacological treatments may promote MM-EV immune activating effect. Indeed, treatment with melphalan increases the capability of MM cells to release exosomes that can activate the NF-κB pathway in NK cells, inducing them to produce IFN-γ [81]. Interestingly, upon stimulation with doxorubicin and melphalan, MM cells and MM-derived exosomes show increased expression of the IL-15/IL15RA complex allowing the functional trans-presentation of IL-15 to neighboring NK cells, with the consequent induction of their activation and proliferation [83].

EVs produced in the tumor microenvironment have been reported to hamper the immune surveillance and play a key role in tumor progression [97]. According to recent studies, immunosuppressive MDSCs accumulate in the BM in the early stages as one of the most prominent immune populations involved in MM progression by suppressing T cell activation and inducing MM cell survival [98]. As a consequence, high levels of MDSC activity promote tumor development [99]. Wang and colleagues demonstrated that MM-MSC-exosomes contribute an important immunosuppressive effect by activating MDSCs [42]. BM MSC-exosomes induce MDSC expansion and survival by activating STAT3 and STAT1 pathways and increasing the anti-apoptotic proteins Bcl-X_L_ and Mcl-1 [42]. Moreover, on a functional level, these exosomes enhance MDSC production of inducible nitric oxide and the suppression of T cell activation (Figure 5B) [42].

Although studying the complex role of EVs in the modulation of immune responses during MM has provided the first exciting results, in order to understand which elements affect EV immune suppressive or immune stimulating behavior, it is crucial to expand the knowledge on the effects of EVs on other immune cell players in the context of this disease, including NK cells, cytotoxic T lymphocytes, T regulatory cells, Th17 cells, dendritic cells, tumor associated macrophages, granulocytes, etc. [100].

Overall, the reported lines of evidence demonstrate how EV contribution is crucial during all the different steps of MM progression, and provide the rationale for further studies aimed at achieving a better understanding of the dynamics involving EV-mediated communication with the premetastatic niche, and during the settlement and growth of new bone lesions. This will allow us to define the molecular basis underlying the crosstalk between MM cells, the surrounding microenvironment, and the metastatic niche, to identify new molecular therapeutic targets in order to improve the final outcome of this disease.

## Figures and Tables

**Figure 1 ijms-20-03236-f001:**
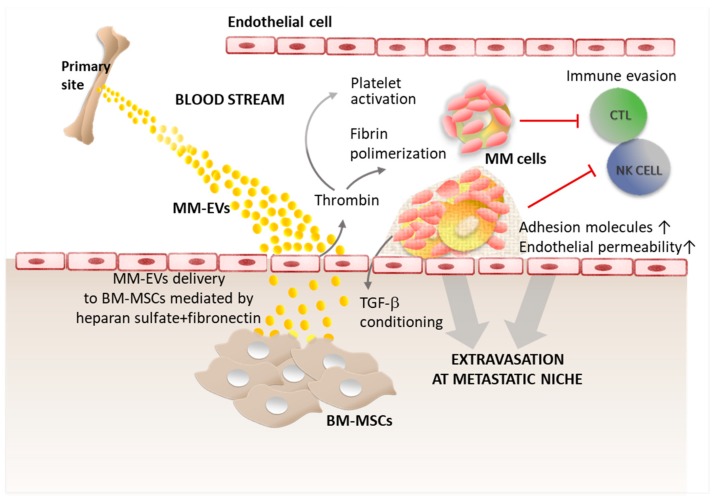
EVs support myeloma cell journey in the blood stream and can be delivered at the premetastatic niche. See the text for details.

**Figure 2 ijms-20-03236-f002:**
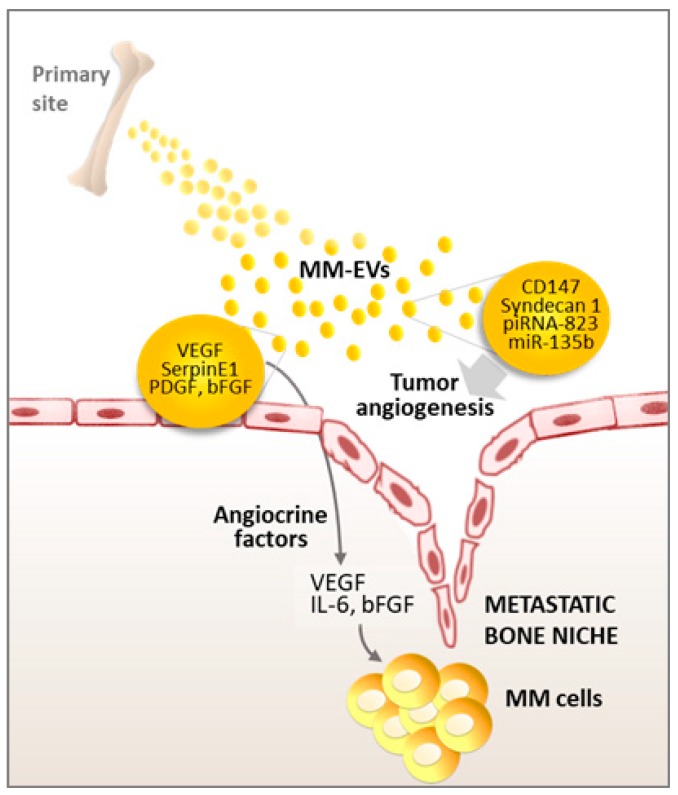
EVs promote angiogenesis and the release of angiocrine factors from endothelial cells. Details in the text.

**Figure 3 ijms-20-03236-f003:**
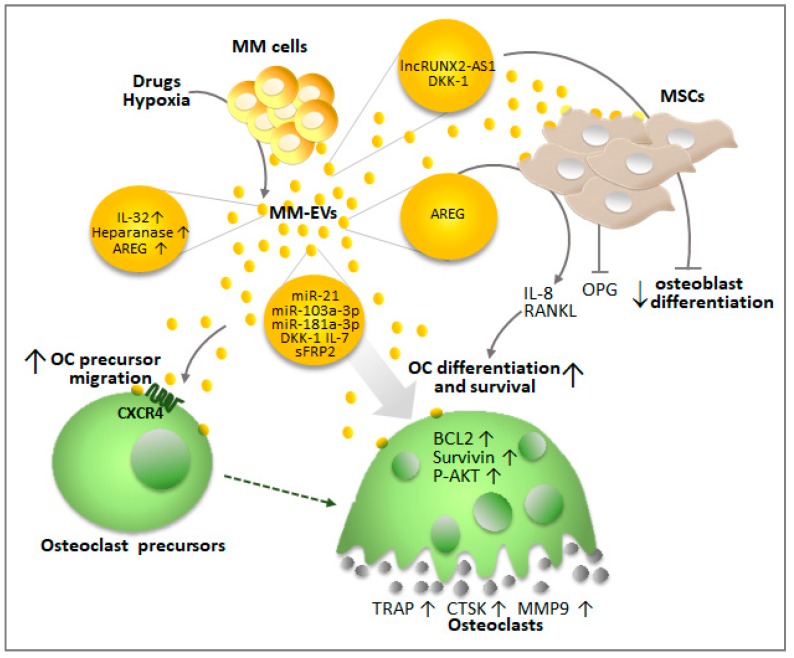
MM-EVs contribute to OC precursor migration, survival, differentiation, and osteolysis. They also inhibit osteoblast differentiation, either directly (through proteins, miRNA and lncRNAs) or by stimulating BM-MSCs. Details in the text.

**Figure 4 ijms-20-03236-f004:**
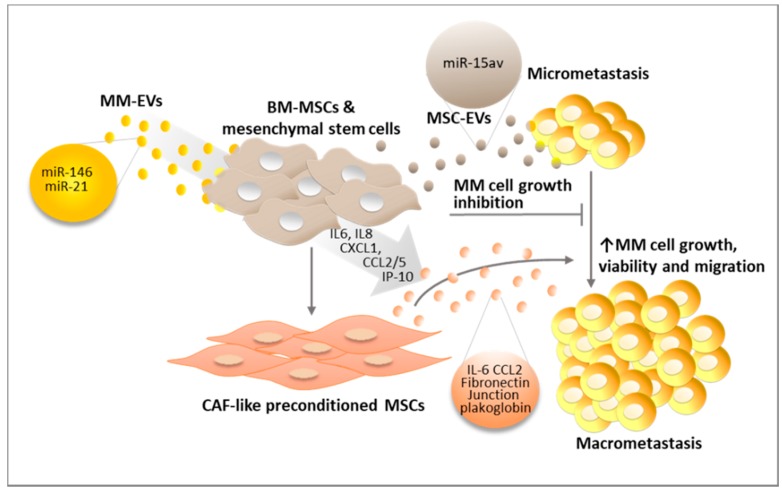
MM-EVs condition BM-MSCs switching their behavior from MM cell growth inhibition (dormancy) to support of tumor cell growth, thereby promoting macrometastasis formation. The role of MSC-EVs and MM-MSC-EVs is shown. Details in the text.

**Figure 5 ijms-20-03236-f005:**
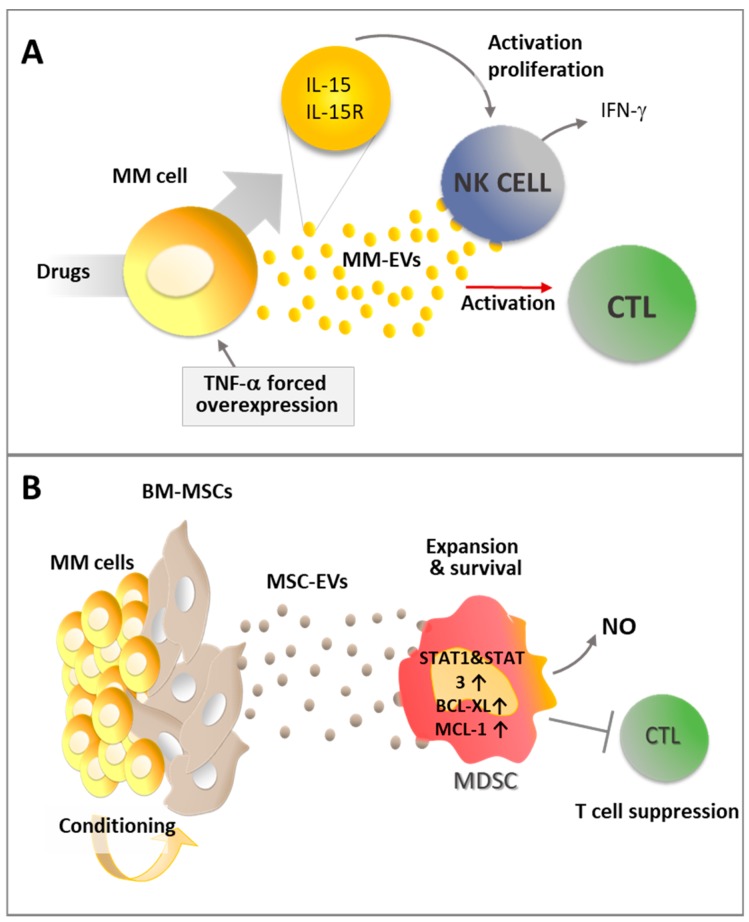
(**A**) MM-EV immune activating effect may be increased by pharmacological treatments and inflammatory stimuli; and (**B**) MM cells may condition BM-MSCs to release immune suppressive EVs that enhance MDSC activity and promote CD8^+^ cytotoxic T lymphocyte (CTL) suppression. Details in the text.

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
