# Peer review of "Extracellular Vesicles Enhance Multiple Myeloma Metastatic Dissemination"

_ijms, 2019, doi:10.3390/ijms20133236_

Round 1
Reviewer 1 Report
The manuscript by Colombo et al., entitled “Extracellular vesicles enhance multiple myeloma widespread” is a review on the current status of research into EVs in the context of multiple myeloma. The review is easy to read, informative and this reviewer especially likes the figures as they are very descriptive and easy to follow. The review is appropriate for IJMS and also for the Special Issue “Extracellular Vesicles and Metastatic Niche”. I would suggest changing the title as it isn’t clear what is actually widespread. There are also many other cases of grammatical errors, mostly plural versus singular English verb tenses. I corrected most, which are listed below. The review would also benefit from a concluding paragraph summarizing the field.
Here are listed the minor grammatical changes (or suggested changes listed):
Line 40 one sentence should not be a paragraph
A sentence should be present that states that given the current methodologies one cannot distinguish between exosomes, microvesicles and apoptotic bodies hence the term extracellular vesicles is used.
Line 83 “to different” not indifferent
Line 91 MM cell not MM cells
Is line 95 a paragraph? A paragraph should not be one sentence.
99 MM cell dissemination and bone lesion
Line 43, 103, 104, 107, 152, 194, 197, 397, 410 EVs should be singular EV
Line 111 exosomes-derived
Line 117 Once left the BM., is awkward consider rephrasing
Line 122, 134, 217, 390 should be MM cell
Figure 1. Thrombin is spelled wrong in the figure.
Line 134 sentence is awkward
Line 141 reword to “the ability of EVs”
Line 152 the “Several lines of evidence” sentence is complex and should be split into two sentences.
Line 158, replace “strongly” to “may”, uptake is also dependent on protein-protein interaction (as listed in next paragraph).
Line 165 MM cells
Line 172 the ability of fibronectin, 173 of should be on
183 cancer cell
193 tip cell
195 involved in
200 tube vessel
224 thereby a decreased angiogenic potential
228 space request is awkward
236, 282, 398 researches should be studies; OCs should be singular OC
Figure 3. Since the inhibit sign is on osteoblast differentiation you can omit the “NO”
247 OCs should be singular OC
252 increases should be increase
266 imbalance not unbalance
Figure 4. plakoglobin is spelled wrong
340 the ability of tumor cells
Author Response
The manuscript by Colombo et al., entitled “Extracellular vesicles
enhance multiple myeloma widespread” is a review on the current status
of research into EVs in the context of multiple myeloma. The review is
easy to read, informative and this reviewer especially likes the figures
as they are very descriptive and easy to follow. The review is
appropriate for IJMS and also for the Special Issue “Extracellular
Vesicles and Metastatic Niche”. I would suggest changing the title as
it isn’t clear what is actually widespread. There are also many other
cases of grammatical errors, mostly plural versus singular English verb
tenses. I corrected most, which are listed below. The review would
also benefit from a concluding paragraph summarizing the field.
We thank you for the helpful revision. We have changed the title in "Extracellular vesicles enhance multiple myeloma metastatic dissemination", revised the english language and inserted a concluding paragraph (lines 454-461.).
Question #1
Here are listed the minor grammatical changes (or suggested changes listed):
Line 40 one sentence should not be a paragraph
A sentence should be present that states that given the current methodologies one cannot distinguish between exosomes, microvesicles and apoptotic bodies hence the term extracellular vesicles is used.
Line 83 “to different” not indifferent
Line 91 MM cell not MM cells
Is line 95 a paragraph? A paragraph should not be one sentence.
99 MM cell dissemination and bone lesion
Line 43, 103, 104, 107, 152, 194, 197, 397, 410 EVs should be singular EV
Line 111 exosomes-derived
Line 117 Once left the BM., is awkward consider rephrasing
Line 122, 134, 217, 390 should be MM cell
Figure 1. Thrombin is spelled wrong in the figure.
Line 134 sentence is awkward
Line 141 reword to “the ability of EVs”
Line 152 the “Several lines of evidence” sentence is complex and should be split into two sentences.
Line 158, replace “strongly” to “may”, uptake is also dependent on protein-protein interaction (as listed in next paragraph).
Line 165 MM cells
Line 172 the ability of fibronectin, 173 of should be on
183 cancer cell
193 tip cell
195 involved in
200 tube vessel
224 thereby a decreased angiogenic potential
228 space request is awkward
236, 282, 398 researches should be studies; OCs should be singular OC
Figure 3. Since the inhibit sign is on osteoblast differentiation you can omit the “NO”
247 OCs should be singular OC
252 increases should be increase
266 imbalance not unbalance
Figure 4. plakoglobin is spelled wrong
340 the ability of tumor cells
Answer #1
We apologize for these inconveniences and have revised and corrected the whole manuscript accordingly to the suggestions.
In particular, concerning the required sentence on methodological limitation in studying extracellular vesicles, we have included a sentence in lines 66-68 of the revised manuscript.
Reviewer 2 Report
This is a well written review, which provides an excellent and updated overview of the role of extracellular vesicles in the context of multiple myeloma.
Minor points:
-The authors could update mentioned literature through citing the review by Raimondi L et al., on the role of miRNAs as mediators of the cross talk between myeloma cells and the tumor microenvironment (Biomed Res international 2016).
-On the light of recent literature which supports the pivotal role of lncRNAs in MM (Amodio et al., J Hematol Oncol 2018), the authors are invited to address the role of these important and vesicle-released regulators by tumor or stromal cells.
Author Response
This is a well written review, which provides an excellent and updated overview of the role of extracellular vesicles in the context of multiple myeloma.
Minor points:
-The authors could update mentioned literature through citing the review by Raimondi L et al., on the role of miRNAs as mediators of the cross talk between myeloma cells and the tumor microenvironment (Biomed Res international 2016).
-On the light of recent literature which supports the pivotal role of lncRNAs in MM (Amodio et al., J Hematol Oncol 2018), the authors are invited to address the role of these important and vesicle-released regulators by tumor or stromal cells.
Answer
We have included the mentioned papers among the references. The role of lncRNAs in MM-secreted exosomes unfortunately has not been deeply studied up to now, so we included a brief discussion on this topic (lines 293-297).
Reviewer 3 Report
This review article describes the roles of extracellular vesicles in biology of multiple myeloma (MM). This paper is well organized and will provide several insights into the understanding of therapies against MM. Several modifications will make this paper more beneficial to the readers of International Journal of Molecular Sciences. The specific points are as follow.
Specific points
1. Page 10, line 224-226. The sentence, “suggesting a further EV-related mechanism though which these two chemotherapeutics contrast MM progression.”, is not clear. It will be better to modify the sentence.
2. Page 11, line 250-256. This review describes that bortezomib, carfilzomib and melphalan increases the levels of heparanase in MM-derived exosomes and that high level of heparanase stimulates systemic osteoclastogenesis and osteolysis. In general, bortezomib is supposed to improve the osteolytic lesions in patients with MM. The authors should explain the discrepancy.
Author Response
This review article describes the roles of extracellular vesicles in biology of multiple myeloma (MM). This paper is well organized and will provide several insights into the understanding of therapies against MM. Several modifications will make this paper more beneficial to the readers of International Journal of Molecular Sciences. The specific points are as follow.
Specific points
Question #1
1. Page 10, line 224-226. The sentence, “suggesting a further EV-related mechanism though which these two chemotherapeutics contrast MM progression.”, is not clear. It will be better to modify the sentence.
Answer #1
We apologize for the inconvenience and modified the sentence (now at lines 230-232)
Question #2
2. Page 11, line 250-256. This review describes that bortezomib, carfilzomib and melphalan increases the levels of heparanase in MM-derived exosomes and that high level of heparanase stimulates systemic osteoclastogenesis and osteolysis. In general, bortezomib is supposed to improve the osteolytic lesions in patients with MM. The authors should explain the discrepancy.
Answer #2
We have proposed a possible explanation for this discrepancy in lines 265-268.
Reviewer 4 Report
General comments:
The review by Colombo and coworkers provides an overview on the role of extracellular vesicles (EVs) in myeloma with specific emphasis on myeloma cell spread, metastasis and bone disease.
In addition, the authors describe how EVs in MM promote anti-tumor immunity but also influence MM immune escape, a dual effect which is modulated by certain treatments.
In summary, the manuscript needs some restructuring, the contents of the chapters overlap to a large extent, which results in some degree of redundancy. For example, tumor cell dissemination (chapter 2) is kind of the same as myeloma cell journey to the metastatic site (chapter 3). I would propose to dedicate chapter 2 solely to multiple myeloma and bone disease/metastasis in a more general way.
The authors should concentrate on myeloma, findings in solid tumors or leukemias rather lead to confusion. In some aspects, such as bone disease, solid tumors differ from myeloma, other findings from leukemia have not been reported for myeloma.
Language and style could be improved. The sentences tend to be very long and are difficult to read. By reducing redundancy, the manuscript could be shortened.
Specific points:
Chapter 1:
The authors describe that the exosomal cargo reflects, without overlapping, the composition of the producing cells. Later it is stated that the content of the EVs partially reflects the cellular origin and that tumor-derived EVs share proteins with other EVs. This is not clear, what do the authors mean with “without overlapping”? The terms MVB and ILV are used equally, are these different vesicles or identical? The enumeration of proteins in the EVs is not reflected by reference #6. ESCRT (page 3, line 53) needs to be explained. Line 59, grammars: … it seems clear that it involves ….
Chapter 2:
First paragraph on MM: at least one reference should be provided.
Line 97: “by exosomes derived from lung tropic metastatic models”, this part of the sentence is unclear, please rephrase or delete. Line 100 ff: it should be mentioned that this was a finding in myeloma patients.
Chapter 3:
Line 128: “The interaction between platelet/fibrin and tumor cells promotes…”
Chapter 4:
Title: Uptake of EVs and cargo delivery to ….
The part on heparin sulfate proteoglycans and syndecan-1 could be shortened and more concise.
Line 172ff: “… based on the ability of fibronectin to bind to heparin sulfate molecules on membrane surfaces.”
Chapter 5:
The first part is very general, what is related to myeloma, has Notch ligand been found in MM-EVs? Line 191 “tumor-derived EVs”, which tumors?
Line 190: do the authors mean “macrometastases”?
Line 203: signal transducer and activator of transcription
Chapter 6:
Ref. 52 and 53: Here it is shown that EVs actually impair osteoclast differentiation. This is in line with an osteoblastic phenotype of skeletal metastases in patients with metastatic prostate cancer. Ref. 53 goes into the same direction, so with this regard prostate cancer and myeloma differ from each other. This sentence could be deleted.
Chapter 7:
This chapter is very long, shows some redundancy with regard to ref. 12, and contains aspects that were found in breast cancer, not myeloma. Despite their tumor suppressor role, have the different miRNAs, e.g. miR-23b, been identified in EVs of MSC, as shown in figure 4?
Chapter 8:
Again, the focus should be on myeloma,
Fig. 5 and line 385: what do the authors mean by “EVs produced by MM cells overexpressing TNF”? That EVs overexpress TNFalpha or MM cells? In the figure “TNF-alpha induced overexpression”: overexpression of what?
Line 409: which other immune cell players do the authors mean?
CTL and MDSC are not explained.
The last sentence should be not be separated?
Figure 3: OC precursor migration: an upwards pointing arrow might be good (á), also behind “OC differentiation and survival”; one could do the same for osteoblast differentiation with an upward pointing arrow. Osteoclasts (plural).
Figure 4: “… MM cell growth inhibition …. to tumor cell growth support …”
Miscellaneous/grammars:
References need reformatting.
The genitive plural form has an apostrophe at the end: EVs’ MVBs’, etc. Stylistically better would be “The origin of EVs”, and so on. In many cases, the singular form should be used, e. g. line 91: “… MM cell homing”, “tumor cell settlement” (line 93), “BM microvessel density” (line 219).
Line 94: formation of new tumor lesions
Ref. 58 is dispensable
Line 246: “AKT phosphorylation” instead of “ratio pAKT/AKT tot”;
Line 247: bone resorbing activity
Line 261: “osteoclastogenic” or “involved in osteoclastogenesis”, either one should be deleted
Line 263: why is EGFR fully written here?
Line 265: cytokine (not cytokines)
Line 270: “several” instead of “other” groups
Reference 11 and 67 are identical.
Line 290: has sFRP2 been explained before?
Line 300: Wang and coworkers
Author Response
The review by Colombo and coworkers provides an overview on the role of extracellular vesicles (EVs) in myeloma with specific emphasis on myeloma cell spread, metastasis and bone disease.
In
addition, the authors describe how EVs in MM promote anti-tumor
immunity but also influence MM immune escape, a dual effect which is
modulated by certain treatments.
Question #1
In summary, the manuscript needs some restructuring, the contents of the chapters overlap to a large extent, which results in some degree of redundancy. For example, tumor cell dissemination (chapter 2) is kind of the same as myeloma cell journey to the metastatic site (chapter 3). I would propose to dedicate chapter 2 solely to multiple myeloma and bone disease/metastasis in a more general way.
The authors should concentrate on myeloma, findings in solid tumors or leukemias rather lead to confusion. In some aspects, such as bone disease, solid tumors differ from myeloma, other findings from leukemia have not been reported for myeloma.
Language and style could be improved. The sentences tend to be very long and are difficult to read. By reducing redundancy, the manuscript could be shortened.
Answer #1
We thank the reviewer for the suggestions, we have tryed to address all the indicated points, avoiding to refer to other types of tumors when not strictly necessary. Moreover, english style and sentence structure have been revised in the whole manuscript and chapter 2 and 3 have been shorted and revised to avoid redundancies.
Question #2
Specific points:
Chapter 1:
The authors describe that the exosomal cargo reflects, without overlapping, the composition of the producing cells. Later it is stated that the content of the EVs partially reflects the cellular origin and that tumor-derived EVs share proteins with other EVs. This is not clear, what do the authors mean with “without overlapping”? We have reformulated the phrase at lines 57-60 of the revised manuscript.
The terms MVB and ILV are used equally, are these different vesicles or identical? We have tried to be more clear by explaining that ILV are contained within MVBs. Please see at lines 52-54.
The enumeration of proteins in the EVs is not reflected by reference #6. We apologize and have corrected the error.
ESCRT (page 3, line 53) needs to be explained. We have explained the acronym
Line 59, grammars: … it seems clear that it involves …. We apologize and have corrected the error.
Chapter 2:
First paragraph on MM: at least one reference should be provided. We have provided the appropriate reference
Line 97: “by exosomes derived from lung tropic metastatic models”, this part of the sentence is unclear, please rephrase or delete. Since not completely pertinent to MM, we have deleted the phrase
Line 100 ff: it should be mentioned that this was a finding in myeloma patients. Mentioned at line 107
Chapter 3:
Line 128: “The interaction between platelet/fibrin and tumor cells promotes…” Corrected
Chapter 4:
Title: Uptake of EVs and cargo delivery to …. Modified
The part on heparin sulfate proteoglycans and syndecan-1 could be shortened and more concise. We have shortened this part (lines 173-184).
Line 172ff: “… based on the ability of fibronectin to bind to heparin sulfate molecules on membrane surfaces.” Corrected
Chapter 5:
The first part is very general, what is related to myeloma, has Notch ligand been found in MM-EVs? Line 191 “tumor-derived EVs”, which tumors? We agree and has revised and shortened this part, reducing general concepts.
Line 190: do the authors mean “macrometastases”? Yes, we have corrected the word, thank you
Line 203: signal transducer and activator of transcription Sorry for the inconvenient, we have corrected
Chapter 6:
Ref. 52 and 53: Here it is shown that EVs actually impair osteoclast differentiation. This is in line with an osteoblastic phenotype of skeletal metastases in patients with metastatic prostate cancer. Ref. 53 goes into the same direction, so with this regard prostate cancer and myeloma differ from each other. This sentence could be deleted. We apologize and have deleted the sentence.
Chapter 7:
This chapter is very long, shows some redundancy with regard to ref. 12, and contains aspects that were found in breast cancer, not myeloma. Despite their tumor suppressor role, have the different miRNAs, e.g. miR-23b, been identified in EVs of MSC, as shown in figure 4? We have tried to be more concise. Additionally we have explained why in order to confirm the tumor suppressive role of MSC-EVs found in MM, we have reported evidence from bone metastatic breast cancer. As a matter of fact, myeloma cells and bone metastatic breast cancer cells share the same MSCs and therefore the same MSC-EVs. Notably, the reported results were obtained using naïve, non-educated MSCs.
MSC-EVs in figure 4 have been reported to play a tumor suppressive role reflecting the presence of tumor-suppressor miRNA (i.e. miR-15av) found in studies on multiple myeloma (Roccaro et al. The Journal of clinical investigation 2013; 123:1542-55.). As anticipated, miR-23b carried by normal MSC-EVs was shown to play a tumor suppressor effect on bone metastatic breast cancer cells (Lim et al., Cancer research 2011, 71, 1550-60; Ono, et al. Science signaling 2014, 7, (332), ra63).
Chapter 8:
Again, the focus should be on myeloma, We have significantly reduced the part concerning other tumors
Fig. 5 and line 385: what do the authors mean by “EVs produced by MM cells overexpressing TNF”? That EVs overexpress TNFalpha or MM cells? In the figure “TNF-alpha induced overexpression”: overexpression of what? We have clarified the meaning in the text at line 414.
Line 409: which other immune cell players do the authors mean? We have inserted the main immune cell players in MM
CTL and MDSC are not explained. We have explained CTL at line 127, while explanation MDSC was already present at line 242 (232 in the original version).
The last sentence should be not be separated? We have changed the final part.
Figure 3: OC precursor migration: an upwards pointing arrow might be good (á), also behind “OC differentiation and survival”; one could do the same for osteoblast differentiation with an upward pointing arrow. Osteoclasts (plural). Arrows added and osteoclasts corrected
Figure 4: “… MM cell growth inhibition …. to tumor cell growth support …” Done
Miscellaneous/grammars:
References need reformatting.
The genitive plural form has an apostrophe at the end: EVs’ MVBs’, etc. Stylistically better would be “The origin of EVs”, and so on. In many cases, the singular form should be used, e. g. line 91: “… MM cell homing”, “tumor cell settlement” (line 93), “BM microvessel density” (line 219).
Line 94: formation of new tumor lesions
Ref. 58 is dispensable
Line 246: “AKT phosphorylation” instead of “ratio pAKT/AKT tot”;
Line 247: bone resorbing activity
Line 261: “osteoclastogenic” or “involved in osteoclastogenesis”, either one should be deleted
Line 263: why is EGFR fully written here?
Line 265: cytokine (not cytokines)
Line 270: “several” instead of “other” groups
Reference 11 and 67 are identical.
Line 290: has sFRP2 been explained before?
Line 300: Wang and coworkers
Answer
We apologize for the inconvenience and have revised the manuscript accordingly to the raised concerns.
Round 2
Reviewer 3 Report
Lines 230-232
I don't think "contrast" is suitable in this sentence.
Page 11, line 250-256, "which eventually can contribute to bone disease worsening and to patients’ relapse." The authors does not explain how a burst of MM-EVs contributes to patients’ relapse. They should explain this by citing the article by Abe M, et al, "Osteoclasts enhance myeloma cell growth and survival via cell-cell contact: a vicious cycle between bone destruction
and myeloma expansion. Blood. 2004;104(8):2484–2491."
Author Response
We appreciate reviewer’s suggestions and have modified the first sentence (see line 212) and included the mentioned paper (see lines 245-247).
Reviewer 4 Report
General comments:
In summary it is a good overview on the current knowledge of extracellular vesicles in multiple myeloma. Most of the issues had been addressed by the authors, however, some additional points came up. Chapters 7 and 8 need some restructuring to reduce redundancy. A number of points remain, which are related to language and grammars. The reviewer tried to list all of them and tried to improve readability, if the authors agree with the suggestions.
Chapter 1:
Reference 7 should be exchanged, it is not really indicative for syndecan-1 expression on myeloma cells.
Chapter 3:
Line 110: After “hypothesis” starting a new sentence would improve readability.
Line 113: “Further evidence ….” This sentence is dispensable.
Line 115: “in vivo” occurs twice
Chapter 4:
“Thanks to their composition” (line 143) and “pH” (line 168) are dispensable.
Line 179 ff: for more clarity, do the authors mean that syndecan-1 expressed on exosomes favors uptake by BM-MSC?
Chapter 6:
Figure 3/Legend: Better would be “MM-EVs contribute to …. either directly ….”; “OC precursor migration” (figure legend and figure)
Line 259: “also in this case” is dispensable
Line 262: “Heparanase activates the ERK pathway in MM cells which results in the release of ….”
Line 264: “high levels of heparanase stimulate ….”
Line 269: “interleukin” is fully written out here, although IL-6 occurred long before
Line 276: “…. differention into mature OCs. Moreover, AREG ….”
Line 284 ff: “…. resulting in ….” This second part of the sentence is dispensable.
Line 286: “MM-EVs in the murine 5TGM1 model….”
Line 293: “Of note, increasing evidences support….” The first part of the sentence (“even if the majority …” is dispensable.
Line 301: “are part of…” This part of the sentence is redundant and can be deleted.
Chapter 7:
Line 322: “The underlying mechanism….” This sentence is redundant and can be deleted.
Line 327 ff.: This part on MSC-EVs needs restructuring, the sentences are too long, and redundancy should be avoided. For example, “promote MM cell proliferation and dissemination, due to …. and fibronectin (ref. 15).” End of the sentence, and the next sentence can be deleted. A strategy for restructuring could be to start from the data on tumor suppression and then move forward to support of tumor growth (as in the figure legend).
Figure 4: The arrow indicating the cytokine release by BM cells (IL-6, and so on) points to CAF-like preconditioned MSCs, shouldn’t it point directly to the MM cells according to the text on line 325? MSC-EVs are indicated in the figure, but not the MM-MSC-EVs.
miR-15av in MSC-EVs, a finding in myeloma, should be included in the figure.
Is it pakoglobin (figure) or plakoglobin (text)?
Figure legend“… to switch their behavior …. thereby promoting macrometastasis formation”
Chapter 8:
Line 396: “…, and by carrying cytokines and chemokines”.
Line 405, “in the paragraphs below”: what about the references given above, are these examples without direct involvement? Examples of tumor antigen expression on MM-EVs are missing (although having been the subject of a different chapter already).
Line 418 ff.: The sentence should be divided into several ones, shortened and rephrased ( “trigger…by….via…. induced by” is too nested).
Spelling/grammars:
References: the format is not uniform, journal titles and page numbers need to be checked.
“Premetatastatic” is predominantly used in the text, sometimes “pre-metastatic”, this should be checked
The apostrophe is still missing in many cases:
- line 27: EV’s or EVs’ (if singular or plural)
- lines 41, 60 (better: Fusion of MVBs with…), 62, 73 (better: the content of EVs), 81 (Multiple myeloma cell dissemination), 148 (EVs’ or “Uptake of EVs”), 373 (EVs’ cytokine content)
The singular form should be used in:
- line 26 (MM cell engraftment)
- line 28 (mesenchymal cell support)
- line 61 (exosome release)
- lines 116 and 122 (MM cell dissemination)
- line 141 (platelet activation)
- line 238 (tumor cell settlement)
- line 244 (OC lytic activity)
- Fig. 3: OC precursor migration
- line 277 (MSC osteogenic differentiation)
- line 279: MSC-mediated release
- line 288: OB differentiation
- line 313: BM-MSC ability
- line 317: BM-MSC viability; and increase (instead of increased)
- line 321: should be “… stimulate cytokine production”
- line 322: transformation into …
- line 337: miRNA content
- line 345: MSC education
- line 373: NK cell cytotoxicity
- line 402: BM-MSC mediated release
- Fig. 5 legend: MDSC activity, CTL suppression
- line 430: MDSC activity
- line 433/435: MDSC expansion, MDSC production
Line 65: involves different components … (delete “of”)
Line 78: “Additionally, …” should be a new sentence.
Line 79: syndecan-1/CD138
Line 83: “MM mostly relies on the tumor microenvironment …” and “The bone marrow …”
Line 85: comma after “Here”
Line 120: different instead of differently
Line 121: EVs’ (plural)
Line 135: tumor cells
Line 143: increased instead of increase
Line 172: cell surface (without hyphen)
Line 220: ICAM-1
Line 229: “It remains to be investigated if….” and “premetastatic”
Line 233: “and thereby have a decreased angiogenic potential”. New sentence: This suggests, that….
Line 247: Zhang et al. who have ….
Line 255: survivin
Line 307: secreted
Line 321: cytokine production
Line 322: transformation into
Line 375: T helper 17 cells
Line 378: an immunosuppressive
Line 397: have been reported….
Line 404: effects
Line 407: The immunogeneic activity of MM-EVs can be
Line 458: basis
Fig. 2: from endothelial cells
Author Response
We thank the reviewer for the careful reading of the manuscript and appreciate his/her suggestions. We have tryed to address all the indicated points and we have modified Chapters7 and 8 in order to reduce redundancy.
Round 3
Reviewer 4 Report
The authors have addressed all points. Due to the revision of chapter 7 and chapter 8, a couple of minor changes remain to be done.
Specifically, these are:
Figure 3/Legend: Fullstop at the end.
Line 77: syndecan-1 (with a hyphen)
Line 241: “high levels of heparanase stimulate ….(not stimulates)”
Line 288: OB differentiation (or OBs’)
Lines 288 and 293: “… micrometastatic growth…”
Line 295: “Recent work indicates ….”
Line 300: “… the same BM microenvironment as in MM …”
Line 303: “miR-23b … is able to induce …”
Line 368: “… or by suppressing CTL response ….”
Line 383/408: should read “On the one hand” and “on the other hand”
Line 387: “In MM” can be deleted
Line 388: do the authors mean “survivin”? APCs should be fully written
Line 390: represents
Line 392: do the authors mean that MM-EVs from myeloma patients express CD38?
Line 399: “subcutaneous challenging of mice”
Line 404: “Interestingly, upon….” should be a separate sentence.
Line 405: increased expression
Line 410: “Concerning ….” Actually this sentence is redundant and can be deleted.
Line 419: at a functional level
Line 422 ff.: “modulation of the immune response” instead of “anti-tumor immune response”; “in order to understand …”; “to expand the knowledge on the effects of EVs on other immune cell players ….
Line 429: … lines of evidences show how ….”
Reference reformatting, in particular with regard to the journal names, will probably need to be done by the editorial office.